# Drug Efflux Pump Inhibitors: A Promising Approach to Counter Multidrug Resistance in Gram-Negative Pathogens by Targeting AcrB Protein from AcrAB-TolC Multidrug Efflux Pump from *Escherichia coli*

**DOI:** 10.3390/biology11091328

**Published:** 2022-09-08

**Authors:** Rawaf Alenazy

**Affiliations:** Department of Medical Laboratory, College of Applied Medical Sciences-Shaqra, Shaqra University, Shaqra 11961, Saudi Arabia; ralenazy@su.edu.sa

**Keywords:** multidrug resistance, drug efflux pumps inhibitors, mechanisms of antibiotic resistance, superfamily efflux pump RND, AcrAB-TolC drug efflux complex, AcrB protein, hydrophobic trap

## Abstract

**Simple Summary:**

Multidrug-resistant bacterial infections, especially that caused by Gram-negative bacteria, have posed serious health issues worldwide. Bacteria have different mechanisms that can confer multidrug resistance to bacteria, among these mechanisms are drug efflux pumps that play the main role in conferring multidrug resistance by recognizing then expelling a wide range of compounds, especially antibiotics, and reducing their concentration to sub-toxic levels. Small molecule inhibitors that target drug efflux pumps especially the AcrAB-TolC multidrug efflux pump, from *E. coli*, appear as a new promising and attractive approach that could increase the required accumulation of antimicrobials to eliminate bacteria as well as leading to reverse antibiotic resistance and prevent the development of resistance in clinically relevant bacterial pathogens and enhances the activity of antibiotics or prolong their effectiveness.

**Abstract:**

Infections caused by multidrug resistance (MDR) of Gram-negative bacteria have become one of the most severe public health problems worldwide. The main mechanism that confers MDR to bacteria is drug efflux pumps, as they expel a wide range of compounds, especially antibiotics. Among the different types of drug efflux pumps, the resistance nodulation division (RND) superfamily confers MDR to various Gram-negative bacteria species. The AcrAB-TolC multidrug efflux pump, from *E. coli*, a member of RND, is the best-characterized example and an excellent model for understanding MDR because of an abundance of functional and structural data. Small molecule inhibitors that target the AcrAB-TolC drug efflux pump represent a new solution to reversing MDR in Gram-negative bacteria and restoring the efficacy of various used drugs that are clinically relevant to these pathogens, especially in the high shortage of drugs for multidrug-resistant Gram-negative bacteria. This review will investigate solutions of MDR in Gram-negative bacteria by studying the inhibition of the AcrAB-TolC multidrug efflux pump.

## 1. Introduction

Multidrug-resistant (MDR) bacteria have become a severe threat to global health. Paradoxically, this problem comes at a time when many pharma companies withdrew their research and investments in search of new drugs for MDR due to the lack of high financial benefits. As a result, the MDR issue has become one of the biggest health problems in the world [1,2,3,4]. Drug efflux pumps, which are the most important resistance mechanism, confer MDR to bacteria due to their ability to extrude a wide range of unrelated chemical structures, such as various classes of antimicrobials, dyes, lipids, virulence factors, antiseptics, disinfectants and preservatives [5,6]. For this reason, the latter are known as multi-drug efflux pumps [7,8]. Among the different types of efflux pumps, the resistance nodulation division (RND) superfamily is considered the main drug efflux pumps family, as it confers MDR to various species of Gram-negative bacteria. The RND efflux pump AcrAB-TolC from *E. coli* and its homologues are implicated in MDR in Gram-negative bacteria. The AcrAB-TolC multidrug efflux pump presents an excellent model for understanding MDR due to the availability of its chemical structures and function [9,10,11]. Targeting AcrAB-TolC via small molecule inhibitors appears as a new approach to countering MDR in clinically relevant pathogenic bacteria. Structures of AcrAB-TolC in complex with substrates and inhibitors, using X-ray crystallography and/or cryo-electron microscopy, have significantly contributed to defining the mechanism of substrate binding and understanding the pharmacological inhibition of multidrug efflux pumps. These structural studies have defined an inhibitor binding pocket in the AcrB subunit, known as the hydrophobic trap. This was a significant result that has contributed to the effective development of small-molecule efflux pump inhibitors (EPIs) that target AcrB using a structure-guided design. Most of the efflux pump inhibitors show high synergistic activity with antibiotics and work specifically against Gram-positive bacteria, whereas Gram-negative bacteria inhibitors have not yet been used clinically due to their instability and toxicity [12]. Hence, this review will focus specifically on literature describing the structure and function of the AcrB protein from AcrAB-TolC, aiming to discover how we can inhibit its function via using small molecule inhibitors to reverse antimicrobial resistance by re-sensitizing microorganisms to currently used antibiotics where resistance has developed.

### 1.1. Multi-Drug Resistance (MDR)

Pathogenic organisms have mechanisms of resistance to combat the harmful effect of many classes of antimicrobials. MDR is defined as the resistance of pathogenic organisms to one or more antimicrobial agents [13]. Many health organizations such as the World Health Organization (WHO), the Centers for Disease Control and Prevention (CDC) and the European Centre for Disease Prevention and Control (ECDC) have warned that MDR bacterial infections are becoming difficult and costly to treat using expensive medications, and require prolonged treatment regimens [14]. One of the most effective mechanisms that confer MDR to bacteria is multi-drug efflux pumps, which prevent the accumulation of antibiotics inside the cell, thereby reducing their concentration to subtoxic levels [6]. Many countries are facing an increased incidence of infections resulting from MDR Gram-negative bacteria [15], where these organisms caused 80% of all serious bacterial infections [16]. In 2017, the WHO revised its list of MDR pathogens in need of urgent attention. This list indicated that the most critical pathogens are Gram-negative bacteria, such as *Acinetobacter baumannii*, *Pseudomonas aeruginosa* and Enterobacteriaceae, to which which *E. coli* belongs (Figure 1).

### 1.2. Gram-Negative Bacteria Are Difficult to Treat

Gram-positive bacteria have a single cell membrane, while Gram-negative bacteria have a multi-layered envelope, which makes them more difficult to treat due to their multi-layered envelope and essential cellular structures, which have given them high medical importance [2,3,6,16,18]. The multi-layered envelope of Gram-negative bacteria consists of; (i) an outer membrane; (ii) an inner membrane; and (iii) a periplasmic space between them [15]. These multiple membranes provide Gram-negative bacteria with the ability to control the permeability of the harmful substances entering the bacteria, such as toxic substances and antibiotics [2,6,16,18]. Moreover, these membranes provide the Gram-negative bacteria with fundamental resistance against many antibiotics of different chemical compositions [15].

### 1.3. Antibiotic Resistance Mechanisms and MDR

Antibiotic resistance mechanisms in pathogenic bacteria confer the MDR against several different classes of antibiotics. As antibiotics consist of different classes of chemical structures with diverse modes of action, bacteria possess different mechanisms to resist these diverse modes of action [19]. The mechanisms include the inactivation of antibiotics, modification/alteration of drug targets, prevention of antibiotic entry and drug efflux pumps (Figure 2). Drug efflux pumps have gained medical importance due to their function, which is expelling a variety of different and unrelated chemical structures with either antibiotics or antimicrobials before they reach the concentration required to kill the bacteria conferring multi-resistance to more than one class of antibiotics or antimicrobials. In the presence of specific outer risks, especially antibiotics, bacteria resort to employing more than one mechanism at the same time in order to counter these risks and ensure their survival. For instance, Gram-negative *P. aeruginosa* usually utilizes efflux pumps to eliminate the antibiotics, but in specific conditions, it utilizes another mechanism, which is the modification of drug targets [20,21].

### 1.4. Drug Efflux Pumps and Antibiotic Resistance

Efflux pumps are found in all bacteria and are the main mechanism that plays the main role in both antibiotic resistance and MDR [22]. In addition, efflux pumps have “emergency resistance”, which allows the bacteria to develop some mechanisms in order to defend themselves when they are suddenly exposed to antibiotics or toxic agents [3]. In addition to expelling different harmful substances of bacteria, efflux pumps have an essential role in causing complex medical problems such as virulence, pathogenesis and biofilm formation [22,23,24,25,26]. The role of efflux pumps is not limited to the above; they also have an important role in bacterial pathogenicity in various ways, including the secretion of bacterial exotoxins, quorum-sensing molecules and other virulence factors, which earned them medical importance and interest from many doctors and researchers looking for treatment for MDR [25], which means that the inhibition of drug efflux pumps leads to the elimination of many other health problems.

Bacteria have five efflux pump superfamilies, which have been well defined and characterized through microbiological, biochemical and structural studies [27,28]. These families are classified based upon specific properties, such as topology in the membrane, whether they span single or double membranes, the sources of energy, substrate specificity, the similarity of primary sequences and stoichiometry of the multi-subunit complexes [16]. These efflux pump superfamilies are the ATP-binding cassette superfamily (ABC), the small multi-drug resistance family (SMR), the multi-antimicrobial extrusion protein family (MATE) the major facilitator superfamily (MFS) and the resistance–nodulation–division superfamily (RND), which have important properties that confer MDR on Gram-negative bacteria [16].

## 2. Resistance–Nodulation–Division (RND) Superfamily Efflux Pump

RND pumps possess the extraordinary ability to extrude a wide array of substrates, including different classes of antibiotics, such as chloramphenicol, novobiocin, tetracyclines, some β-lactams, fusidic acid and fluoroquinolones. However, substrates are not limited to antibiotics; they also include biocides, detergents, bile salts and metals, as well as compounds synthesized by the bacteria, such as virulence factors and iron-chelating siderophores [29]. About the only feature that these chemicals share is that they are generally amphiphilic [30]. RND transporters capture a wide variety of substrates from the cytoplasm, the outer leaflet of the inner membrane or the periplasm [31]. Hence, the RND transporters can work in concert with other transmembrane transporters that move compounds into these cellular compartments. The RND superfamily consists of multi-protein complexes that span the outer membrane, periplasmic space and inner membrane of Gram-negative bacteria. Accordingly, these protein complexes are larger than many other bacterial trans-membrane proteins, which typically span only a single membrane. The RND superfamily drug transporters consist of three proteins: an inner-membrane protein (IMP), an outer-membrane protein (OMP) and a periplasmic adapter protein (PAP), which connects IMP with OMP (Figure 3). These proteins work as tripartite complex and their function is to recognize and bind substrates from the cytoplasm, the outer leaflet of the inner membrane or the periplasm and the inner membrane then expel these substrates to the outside of the cell via the channel protein which extends to the outer membrane. On the other hand, the other transporters work as individual units in the inner membrane to transfer substrates through the membrane bilayer [16,18].

The *E. coli* AcrAB-TolC drug efflux complex, a member of the RND superfamily, is arguably the best-characterized example, and it is considered an archetypical drug efflux system in Gram-negative bacteria, with homologues found in other species, such as MexAB-OprM in *P. aeruginosa*, MtrCDE in *N. gonorrhoeae* and CmeB in *Campylobacter jejuni* [11].

### 2.1. Structure and Function of E. coli AcrAB-TolC Drug Efflux Pump Complex

The *E. coli* AcrAB-TolC drug efflux pump complex consists of AcrB, which represents IMP and recognizes substrates and utilizes the proton motive force (pmf) for substrate translocation; (ii) TolC, which represents the OMP channel and facilitates the expulsion of substrates from the cell [32]; (iii) AcrA, which represents PAP and links AcrB and TolC and acts to safeguard the transport of the captured substrate across the periplasm (Figure 3) [33,34]. Each subunit of the transporter is an integral part of the efflux machinery. The inhibition of one component can affect the overall function of the multi-subunit efflux pump [33,35,36]. AcrB is the largest protein in the complex. Moreover, AcrB is responsible for multi-drug poly-specificity, as well as for the proton translocation that drives substrate movement [33].

### 2.2. Overview of the Structure of AcrB

More than 20 different structures of *E. coli* AcrB have been solved by either X-ray crystallography or cryo-electron microscopy, making it the most extensively characterized representative of the RND family [37,38]. These include co-complexes of AcrB with a variety of chemically unrelated substrates, knowledge of which aids understanding of the poly-specificity of the transporter [33,39,40,41,42,43,44]. The studies also reveal that conformational changes drive the movement of substrates through the transporter [2,45]. Moreover, molecular simulations have also assisted with understanding the function of the pump and inhibition by small molecules [44,46].

The first crystal structures of AcrB defined all the domains of the protein. A transmembrane region composed of 12 α-helices both anchors the protein in the inner membrane and supports the movement of protons, which is coupled with substrate transport and provides the energy that drives translocation [42,47]. A groove between transmembrane helices 8 and 9 provides one of two well-defined access sites for substrates to enter the multi-protein complex. The remainder of the protein protrudes into the periplasm and is required for substrate binding and docking with periplasmic protein AcrA, as well as TolC that is embedded in the outer membrane. The porter domain of AcrB that spans the periplasm is composed of four subdomains, each containing a common two β-strand–α-helix–β-strand motif. These subdomains are named PN1, PN2, PC1 and PC2. Stacking of the porter subdomains from all three protomers creates a central pore and a long channel for substrate translocation that extends through the homotrimer perpendicular to the membrane. A second substrate access site is located in a cleft formed between the PC1 and PC2 domains. On top of the porter domain are the docking domains DN and DC for binding AcrA and TolC. Antiparallel β-strands create a funnel-like structure at the top of the AcrB necessary to form a tight seal with the base of TolC, preventing substrates from leaking back into the periplasm.

### 2.3. Movement of Substrates through AcrB

The initial crystal structure reported in 2002 by Murakami et al., devoid of any bound Ligands, crystallized as a symmetrical homotrimer [48]. These initial structural data defined the overall architecture of AcrB. Three subsequent studies in 2006 that first reported AcrB in complex with various ligands demonstrated that each of the three protomers adopted a different conformation, resulting in an asymmetrical homotrimer [42,47,49]. This is proposed to be the active state. In this situation, the three alternative conformations have been defined as the loose (L), tight (T) and open (O) states that serve as the access, binding and extrusion conformations (Figure 4) [18,33,50]. As substrates move through the protein, the monomers cycle between L, T, O and back to L. The “L conformer” that is receptive to substrate binding is also considered the resting state [42,43,47]. In the unliganded L state, substrates first enter AcrB from either the inner membrane envelope or the periplasm through one of two well-defined access sites, then diffuse into a proximal binding pocket (PBP). The substrates then move through AcrB towards the distal binding pocket (DBP) located deeper within the porter domain (Figure 3). A conformational change then occurs toward the T state, with the substrates being transferred through the central pore of AcrB toward the entrance of TolC. Substrates are finally expelled from AcrB and into TolC through this functional rotating mechanism that sees the protein transition from the T to O states (Figure 3) [42,47]. In the open O state, a channel is created for the extrusion of the substrate from AcrB and into TolC (Figure 4). During substrate translocation, there are a number of occlusions and constrictions that occur inside the porter domain, driving the unidirectional transport of substrate toward the central funnel. This mechanism has been called the “peristaltic pump mechanism” [47,51,52].

### 2.4. Substrate Entry Sites into the AcrB Promoter

Two primary sites for substrate entry into AcrB that contribute to the poly-specificity of the transporter have been defined. Hydrophobic substrates partitioned in the outer leaflet of the inner membrane can enter via a hydrophobic groove in the transmembrane domain defined by TM8 and TM9 [42,51]. This produces tunnel 1, which extends from the membrane plane at the periplasmic end of the TM8/TM9 groove until it joins tunnel 2. In contrast, compounds localized in the periplasm access through an alternative site located in a cleft between the PC1 and PC2 porter domains, ≈15 Å on top of the membrane plane. This leads into tunnel 2, which extends toward the center of the protein [33].

### 2.5. Substrate Binding Pockets

Tunnels 1 and 2 ultimately connect together, leading to the distal binding pocket buried deeper inside the AcrB protein. The proximal and distal binding pockets are separated by a glycine-rich switch loop that is suggested to assist in the unidirectional movement of substrates (Figure 5) [33]. Structural studies of AcrB reveal that high-molecular-mass drugs such as erythromycin and rifampicin co-crystallize in the PBP, whereas smaller substrates and low-molecular-mass drugs such as minocycline and doxorubicin localize in the deeper DBP (Figure 5) [43,53]. The substrate binding sites of *E. coli* AcrB and the *P. aeruginosa* MexB homologue have similar structures, and thus share broad substrate specificity. The distal binding pocket is a large cavity lined with hydrophobic phenylalanine residues Phe 136, Phe 178, Phe 610, Phe 615, Phe 617, and Phe 628 and polar residues Asn 274 and Gln 176 [54]. Substrates interact with the side chains of these hydrophobic residues via Van der Waals and π-π ring-stacking interactions, and with the polar residues through hydrogen-bonding interactions. The different binding poses adopted by minocycline and doxorubicin highlight the broad substrate specificity of the pump.

The structures also revealed a switch loop (Gln 616—Gly 619) that separates the distal pocket and proximal pocket. It is believed that this loop plays a role in regulating the access of substrates to the distal binding pocket [54]. It is suggested that during translocation, substrates leave the proximal binding pocket and physically contact the switch loop. The flexible loop then moves such that the substrates can now move into the deeper distal binding site. Substrates are not expelled into TolC unless they pass through this switch loop (Figure 5) [33,43,55,56]. Once compounds bind into the distal binding pocket, the protein undergoes a T to O conformational change that sees tunnels 1 and 2 collapse due to substantial reorientation of the porter subdomains and a coil-to-helix transition at the N-terminal end of TM8. A third tunnel is subsequently created in the O state to transport substrates from AcrB towards the TolC extrusion funnel [33]. Quaternary structural changes in AcrB, instigated by the ligand binding, facilitate a change in AcrA that leads to the opening of the sealed base of TolC. Repacking of AcrA plays an important role in preventing the substrates from leaking back into the periplasm by sealing the gaps inside the pump. The TolC channel will not open if the assembly is open to the periplasm. When the substrates are extruded out of the cell, or if there are no substrates inside the pump assembly, the AcrB trimer changes its state from the drug transport confirmation to the resting L state [57].

## 3. Structures of Inhibitors Bound to AcrB—The Hydrophobic Trap

Unlike the symmetrical AcrB homotrimer described above, which readily crystallizes, obtaining co-complexes of the protein with bound substrates or inhibitors has proven more challenging. This is due to (i) the flexible and dynamic multi-site binding mechanism of AcrB, and (ii) multiple conformational changes that drive the movement of substrates during transport [58]. Both processes can make it difficult to grow the well-ordered protein crystals necessary for X-ray diffraction analysis. However, there are now a number of examples of AcrB structures with bound compounds, which aids in understanding the molecular basis of inhibition [33,42,43,44,57,59,60]. The distal binding pocket contains two features that are important for inhibitor binding: (i) a hydrophilic substrate translocation channel that is receptive to multi-drug binding and (ii) a hydrophobic trap adjacent to the distal binding pocket that branches off this channel (Figure 5), which gives it high consideration for the design of small molecule inhibitors that target AcrB. The importance of the trap was first recognized by Nakashima et al. (2013) when they published the first co-crystal structure of the inhibitor, pyridopyrimidine EPI D13-9001 (also known in the literature as ABI-PP), bound to both AcrB from *E. coli* and MexB from *P. aeruginosa* [59]. There is a tight binding interaction between D13-9001 and the phenylalanine-rich hydrophobic trap that prevents the L to T conformational change and, consequently, impedes the functional rotation mechanism. Furthermore, it is suggested that the hydrophilic moiety on D13-9001 blocks substrates from accessing the binding cleft which is located between the PC1 and PC2 domains. A key amino acid in the hydrophobic trap is Phe 178, which plays an important role in inhibitor binding through π-π stacking interactions with aromatic substrates [44,59]. The discovery of this hydrophobic trap is considered an important advance, contributing to the effective development of inhibitors using structure-guided design [61].

## 4. Small Molecules That Inhibit AcrB; Efflux Pump Inhibitors (EPI)

EPIs can be classified based on the sources of their derivation into three categories: plant derivatives, which are the main resource; chemical derivatives; and derivatives from microorganisms [62]. EPIs could be also defined as the small molecules that inhibit drug efflux pumps leading to prevention of the accumulation of antibiotics required to kill bacteria and reducing the resistance and reversal of MDR [22,46]. In AcrAB-TolC, substrates are translocated using the functional rotation mechanism described earlier, which requires all protein subunits to work in concert. Thus, small molecule inhibitors that target one protein in the AcrB homotrimer lead to the inactivation of the entire AcrAB-TolC pump [33]. Small molecules that prevent the action of these multi-drug efflux systems can potentially be administered as an ‘adjuvant or chemosensitizer’ alongside existing antibiotic regimens to improve their efficacy [3,46]. Research efforts are underway to investigate this as an approach to combat MDR due to the paucity of new antibiotics in the pipeline, especially for Gram-negative infections [3,5,63,64,65,66]. However, EPIs have not yet progressed to clinical trials for reasons including high toxicity and poor pharmacology. This makes the discovery of new EPIs that are safer and have greater efficacy an important priority [67].

To identify potential compounds that act as EPIs, small molecules must satisfy the following specific criteria: (i) small molecules must have no antimicrobial activities against bacteria; (ii) the molecules should work specifically on targeted efflux pumps, not on other efflux pumps, as these pumps have a high similarity in their structures, functions and forms, which makes the inhibition of efflux pumps complicated; (iii) the molecules must be safe and non-toxic and must have no bad side effects on bacterial strains [62]. Significant efforts have been made to discover new classes of EPIs. However, as these transporters are poly-specific and have the ability to bind and excrete a wide of unrelated substrates, it has been difficult to identify good preclinical candidates [68].

The first reported EPI was MC-207,110 (Phe-Argβ-napthylamide or PaβN, a chemical derived), described by Lomovskaya et al. in 2001 (Figure 6) [69]. PAβN was shown to inhibit the clinically relevant RND efflux pumps of *P. aeruginosa* and other Gram-negative bacteria such as *E. coli, E. aerogenes, K. pneumoniae* and *S. enterica* [66,69]. Moreover, this peptidomimetic proved to effectively inhibit the expulsion of quinolones, especially levofloxacin and fluoroquinolone [66,69]. PAβN has not been used in the clinical context due to its cytotoxicity [70,71,72]. Subsequently, many different classes of EPIs have been reported. Table 1 summarizes some examples of EPIs from non-plant sources against Gram-negative bacteria, and efflux pump substrates.

Another EPI that has been the subject of drug discovery efforts targeting AcrAB-TolC is the pyranopyridine MBX2319 (Figure 7). This compound is chemical derived and has potent activity against RND efflux pumps of Enterobacteriaceae species [9,103]. MBX2319 was first identified during a chemical screening of over 183,000 compounds for bioactives that act synergistically with ciprofloxacin in a cell-based assay with *E. coli* [9]. MBX2319 at 12.5 µM reduced the MICs of ciprofloxacin, levofloxacin and piperacillin against *E. coli* by 2-, 4- and 8-fold, respectively. Viable cell-counting experiments with *E. coli* AB1157 showed that the bactericidal activity of ciprofloxacin decreased 10,000-fold when used in combination with MBX2319 at 3.13 µM for 4 h compared with antibiotic treatment alone, and it was far superior to PAβN. Furthermore, MBX2319 was not active against *E. coli* strains deficient in AcrAB-TolC and its treatment did cause the accumulation of the AcrAB-TolC fluorescent substrate Hoechst 33342 in *E. coli* AB1157. MBX2319 did not damage the inner or outer bacterial membranes [9]. Together, these data support MBX2319 as a promising preclinical candidate.

As described earlier, a co-complex of MBX2319 bound to AcrBper was solved to define the molecular basis of inhibitor binding. These structural data were then used to design a new series of inhibitors [44]. Large water-filled cavities were observed around the appended benzene moiety of MBX2319, suggesting that chemical elongation of the inhibitors made it possible to realize new derivatives that can form extended bonding interactions and yield greater inhibition. Indeed, the attachment of acetamide or acrylamide groups formed additional hydrogen bonding interactions that improved the potency of the compounds as EPIs and desensitized *E. coli* to ciprofloxacin [44]. Therefore, MBX2319 presents as a promising lead compound that can be further developed through medicinal chemistry.

One of the main categories of EPIs is the botanical kingdom due to its production of a large array of chemicals that can be explored to discover new medicines. Accordingly, phytochemicals have also been investigated as new EPIs. A summary of the reported EPIs from plant origin is shown in Table 2. Plant-derived EPIs can be divided into four main categories; (i) Plant alkaloids such as reserpine, which is a derivative from the roots of Rauwolfia serpentina and showed biological activities targeting the MFS and RND [110]. Like many EPIs, reserpine is not used yet in clinical applications due to its toxicity, especially in the kidneys [62]. (ii) Flavonoids such as Baicalein, the derivative from thyme leaves (Thymus vulgaris). This compound showed an ability to increase the susceptibility of MRSA strains against β lactam and ciprofloxacin antibiotics, but also has not been used clinically yet due to its toxicity [111,112]. (iii) Polyphenols such as Catechin gallates that showed biological activity in reversing the resistance to MRSA and work specifically on NorA drug efflux pump. Due to their toxicity these compounds are not used clinically yet [113]. (v) Phenolic diterpenes such as the derivatives of the herb Rosemary (Rosmarinus officinalis) carnosol and carnosic acid. These compounds can increase the efficacy of some antibiotics such as tetracycline and towards *S. aureus,* working in particular on overexpressing the ABC family complex targeting MsrA and TetK drug efflux pumps in this complex [114]. Likewise, gallotannin (1,2,6-tri-O-galloyl-b-D-glucopyranose), which extracted Terminalia chebula fruits, showed inhibitory activity against the drug efflux pumps of some species of pathogenic *E. coli* by reducing the minimal inhibitory concentration (MIC) of tested antibiotics from 2- to 4-fold by inhibiting the ethidium bromide (EtBr) pump [115]. Along similar lines, catharanthine showed potentiate activities for some antibiotics, such as streptomycin and tetracyclin, against *P. aeruginosa* also reduced MICs and works as EPI [116]). Additionally, some other phytochemicals such as gingerol, capsaicin, resveratrol, catechol and pinene showed inhibitory activities that increase their possibility to be excellent EPIs [117].

As another example, Ohene-Agyei et al. identified five new EPIs of botanical origin by using in silico screening that was specifically intended for inhibitors of AcrB [12]. This included, for stance, the naphthoquinone shikonin, from the traditional Chinese herb *Lithospermum erythrorhizon*, and the antioxidant nordihydroguaiaretic acid (NDGA), from the creosote bush (Figure 8). The antibacterial activities of shikonin and NDGA were tested to determine their ability to function as EPIs. Both compounds showed synergistic activity with a panel of antibiotics, with similar potency for the peptidomimetic PAβN. Furthermore, neither compound disrupted the bacterial outer membrane, which indicated that the observed synergy was not due to a non-specific mechanism such as membrane disruption [12]. However, toxicity against the inner bacterial membrane was not addressed in this study. Importantly, shikonin and NDGA did inhibit the AcrAB-TolC drug efflux pump in whole cell substrate transport assays [12].

## 5. Conclusions

In conclusion, AcrAB-TolC multidrug efflux pump, from *E. coli*, is an excellent model and best-characterized example for understanding MDR in Gram-negative bacteria, so it was studied deeply in this review to know the best way to disable its functions. Among subunits of AcrAB-TolC, AcrB has been given more attention to inhibit it via small molecule inhibitors due to its primary role in multidrug poly-specificity for proton translocation that drives substrate movement, if AcrB is inhibited, entire AcrAB-TolC will inhibited. AcrB has a hydrophobic trap adjacent to the distal binding pocket that branches off this channel playing the leading role in designing small molecule inhibitors that target AcrB. The trap has an importance in the tight binding of the inhibitors with the phenylalanine that is located inside this trap, its inhibition leading to preventing the conformational change and, consequently, impeding the functional rotation mechanism. The hydrophobic trap has a key amino acid which is Phe 178, which plays an important role in inhibitor binding through π-π stacking interactions with aromatic substrates. that play as an important site for inhibitor binding and pharmacological targeting of the efflux machinery using small molecules which will contribute to the effective development of inhibitors using structure-guided design. This review suggests that the chemical elongation and the size of the potential inhibitors should be considered for new inhibitors which leads to realizing new derivatives that can form extended bonding interactions and yield greater inhibition. Importantly, the inhibitors must have a defined mechanism of action, work specifically on the drug efflux pumps, be easily soluble and have cellular permeability able to enter the bacterial cell, have no unwanted off-target effects and don’t disrupt the bacterial outer or inner membranes as well as have no toxic effect either on the mammalian cells. Among the suggestions in this review is that the inhibitors should possess broad-spectrum activity against a wide range of Gram-negative bacteria. This review also recommends a full search of the genome sequence in order to identify the mutations that caused the emergence of multi-drug resistance. The review also recommended facilitating some of the obstacles that stand in the way of using inhibitors as therapeutic aids after ensuring their work and freeing them from potential toxicity.

## 6. Discussion

The World Health Organization (WHO) warned of ‘priority pathogens’, most of which are Gram-negative bacteria, due to the rise of antibiotic resistance and the lack of new products developed to treat the issue [127]. In the past two decades, only a few new antibiotics to combat drug-resistant bacteria have progressed to clinical use [128,129,130]. This is of grave concern, as infections caused by Gram-negative pathogens have high morbidity and mortality rates worldwide [129]. Compared to Gram-positive bacteria, Gram-negative bacteria are tolerant of a wider variety of antimicrobials due to their formidable outer membrane, which acts as a barrier preventing the entry of harmful compounds into the cell, making them difficult to treat [127,131]. Furthermore, Gram-negative pathogens can combat antibiotics and other harmful chemicals through the over-expression of multi-drug efflux pumps [16,22]. Efflux pumps are highly effective at protecting bacteria from chemical assault due to a variety of efflux systems and the drug poly-specificity of these proteins [131]. Gram-negative bacteria contain all the different classes of efflux pumps; however, the resistance–nodulation–division (RND) class, of which the AcrAB-TolC multi-drug efflux pump from *E. coli* and other Enterobacteriaceae is part, is the clinically relevant efflux system. In this class of transporters, AcrAB-TolC is the best-characterized example and is an excellent model for understanding MDR because of an abundance of functional and structural data [9,10,11]. Hence, inhibition of these pumps using inhibitors consider a new promising solution to combat MDR and increases the effectiveness of antibiotics.

Structural biology has played a key role in understanding the function of AcrB. In particular, the discovery of the hydrophobic trap as a site for inhibitor binding has assisted the development of small molecules that specifically target AcrB [123]. The hydrophobic trap that resides adjacent to the DBP (Figure 9) is lined with several aromatic amino acids that play a critical role in poly-substrate binding through hydrophobic and π-π stacking interactions [44,59]. Compounds that bind into this pocket are proposed to either prevent the movement of compounds through the translocation tunnel or prevent the conformational changes that drive substrate translocation [54,61]. Thus, this discovery has greatly contributed to the effective design of inhibitors using structure-guided design [61].

One important suggestion in this review is to design inhibitors that have the ability to bind to the DBP, which leads to specifically targeting the main efflux pump in clinically important Gram-negative bacteria. This approach has the potential advantage of finding new safer preclinical candidates with a defined mechanism of action. This is important, as many EPIs often have issues with toxicity due to unwanted off-target effects. Moreover, it also exploits early stage discovery work that has identified multiple classes of inhibitors, thus expanding our repertoire of compounds that are ready for chemical optimization programs. In order to design small molecule inhibitors that target AcrB, it is necessary to appreciate the role of the distal binding pocket. This pocket contains two features that are important for inhibitor binding: (i) a hydrophilic substrate translocation channel that is receptive to multi-drug binding and (ii) a hydrophobic trap adjacent to the distal binding pocket that branches off this channel. Therefore, an important consideration for future inhibitors design should be (1) the size and length of the inhibitors and (2) whether they can span the full length of the hydrophobic trap for full inhibition.

One main suggested assay for potential EPIs is testing their toxicity on mammalian cells to ensure that they are not toxic to human cells [132]. It is highly desirable to find EPIs that inhibit antibiotic efflux across multiple pathogenic Gram-negative bacteria [133,134]. This review has focused on the AcrAB-TolC multi-drug efflux pump from *E. coli*. We recommend performing many assays in vitro to confirm these bioactivities on a larger panel of bacteria that includes clinical isolates of *E. coli* and antibiotic-resistant strains. Furthermore, as homologues of AcrAB-TolC are also present in other Gram-negative bacteria, such as *K. pneumoniae* and *P. aeruginosa*, the EPIs identified here many have broader activity against these species. However, this needs to be confirmed through empirical laboratory testing. Should strains that are resistant to the EPIs be identified, then whole-genome sequencing should be considered to identify mutations that give rise to the resistance. These mutations may reside in the AcrAB-TolC protein, other efflux transporters or their transcriptional regulator proteins.

There are challenges that prevent the use of EPIs as therapeutic adjuvants such as scientific, administrative and economic challenges. Many of those interested in developing therapeutics against antibiotic resistance tend to focus on chemically modifying the composition of the antibiotics so that they become more effective because of the full knowledge of their chemical composition and mechanism of action. EPIs are considered a new chemical entity (NCE), which entails many strict regulatory, financial and administrative requirements, which makes this task difficult. Moreover, their mechanisms of action and side effects such as problems of toxicity, insolubility and cellular permeability are not fully known yet [75]. Another challenge is related to the nature of the origin of the EPIs. Natural EPIs often have no toxicity problems, but there are issues with their permeation into bacterial cells due to their complex structure and large size. While synthetic EPIs are easy to form and their size is easy to control to fit into cells, they tend to have high potential toxicity in mammalian cells. EPIs are adjuvant therapies which require the presence of another compound, antibiotics, which makes compatibility in the mechanism of action and association another challenge [75].

During the last two decades, EPIs of multi-drug resistance bacteria have been studied intensively by many research groups that are interested in the MDR field, yet no candidates have successfully completed clinical trials, often due to concerns regarding their safety [128,129,130]. Other reasons for this are the downturn in investment by pharmaceutical companies in the clinical development of antibiotics [3,135,136], as well as strict and complicated regulatory requirements [74]. However, as the clinical need to address antibiotic resistance escalates and the efficacy of these existing antibiotic classes decreases, policymakers and pharmaceutical companies will be forced to consider new solutions to combat these infections, such as the development of EPIs.

## Figures and Tables

**Figure 1 biology-11-01328-f001:**
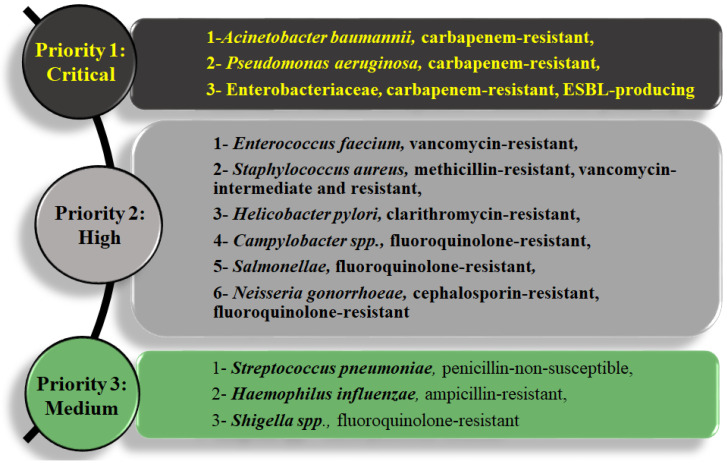
WHO list of priority pathogens in most urgent need for antimicrobial development [17].

**Figure 2 biology-11-01328-f002:**
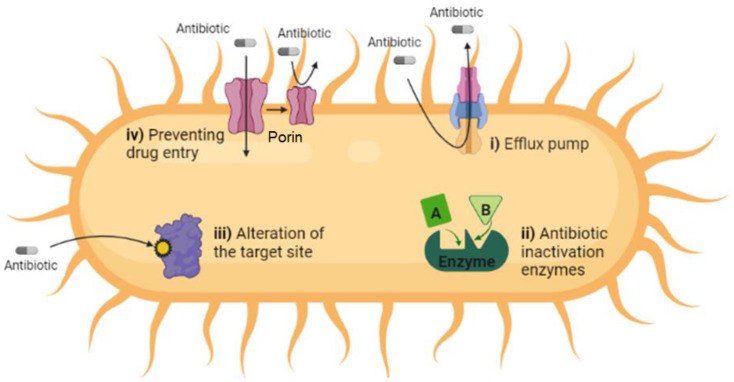
Schematic diagram highlighting the main antimicrobial resistance mechanisms in pathogenic bacteria, which include: (i) drug efflux systems, which expel the antimicrobials outside the bacterial cell and reduce their effectiveness to non-toxic levels. (ii) Antibiotic inactivation enzymes, which modify or destroy the structure of antibiotics. (iii) Alteration of the target site, which usually occurs in the cell envelope via spontaneous mutation via chemical modification of their molecular targets. (iv) Preventing drug entry by modifying the frequency, size and selectivity of porin channels, which are found in the bacterial envelope and play a crucial role in antibiotic entry into the bacterial cell.

**Figure 3 biology-11-01328-f003:**
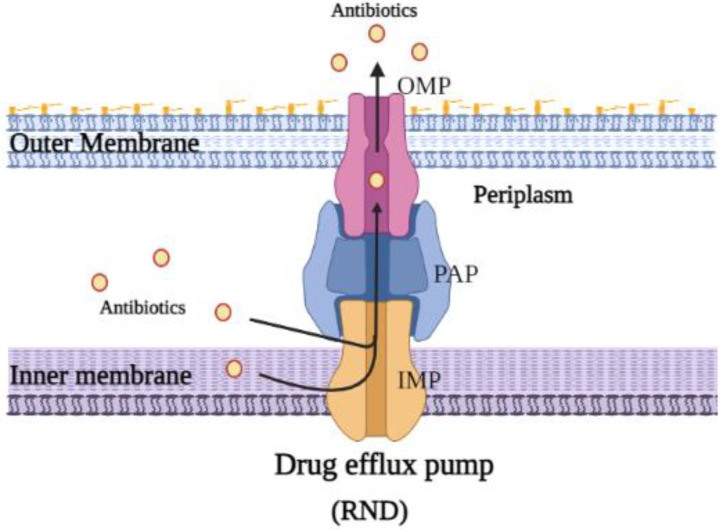
Schematic diagram depicting representative structures of the resistance–nodulation–cell division (RND) family. The RND superfamily are tripartite drug transporters and are formed of three parts; an inner-membrane protein (IMP), an outer-membrane protein (OMP) and a periplasmic adapter protein (PAP), which works as a linker between IMP with OMP.

**Figure 4 biology-11-01328-f004:**
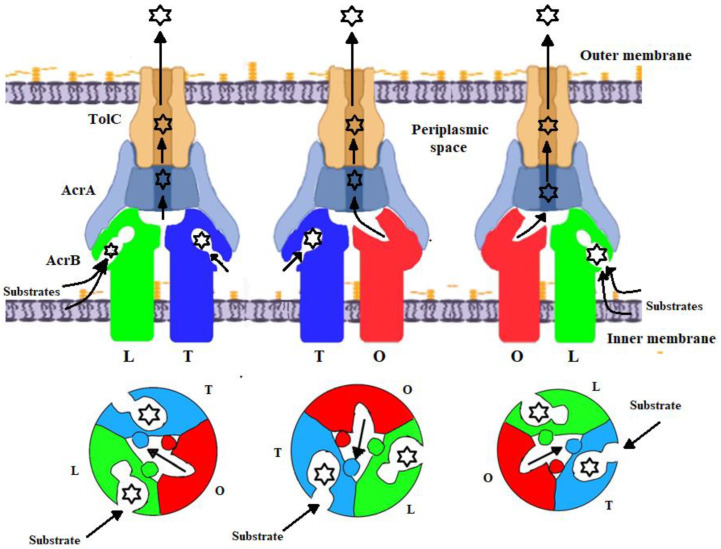
Schematic diagram showing the functional rotational mechanism of the tripartite complex AcrAB-TolC drug efflux pump. The upper and lower panels show the side and horizontal views of the complex, respectively. The three conformations of AcrB, defined as the loose (L: green), tight (T: blue) and open (O: red) are depicted that serve as the substrate access, binding and extrusion states, respectively. AcrA (light blue) and TolC (pale orange) are also shown. Substrates are indicated by black jagged circles. The movement of protons through AcrB is also shown.

**Figure 5 biology-11-01328-f005:**
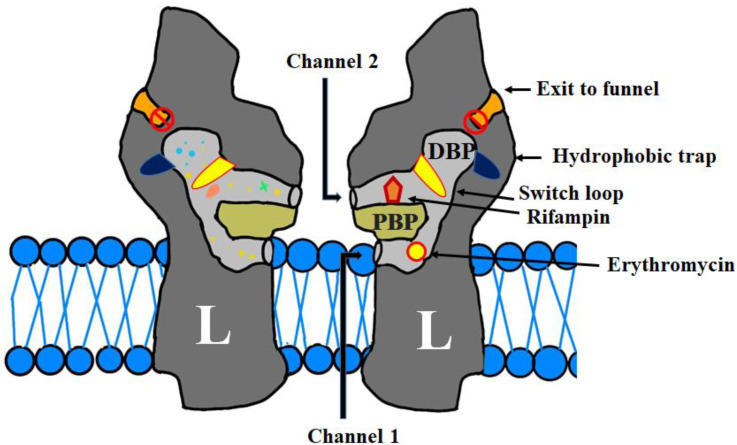
Schematic diagram depicting drug pathways in AcrB and the drug-binding pockets and the two channels that transport substrates are depicted. The periplasmic entry site (channel 2) includes the proximal binding pocket (PBP) in the L protomer, while channel 1 guides substrates from the inner membrane leaflet. The binding sites for rifampicin and erythromycin, co-crystallized in the access pocket in the L monomer, are shown. Minocycline and doxorubicin bind in the deep binding pocket within the T monomer (not shown). The switch loop that separates the proximal and distal binding pockets is also shown. Moreover, the schematic diagram shows the proximal and distal binding pockets of AcrB and a switch loop that separates them, which assists in the unidirectional movement of substrates, whereas smaller substrates and low-molecular-mass drugs localize in the deeper DBP. The two entry channels for substrates connect together leading to the distal binding pocket shown, as is the exit channel that leads from the central cavity to TolC.

**Figure 6 biology-11-01328-f006:**
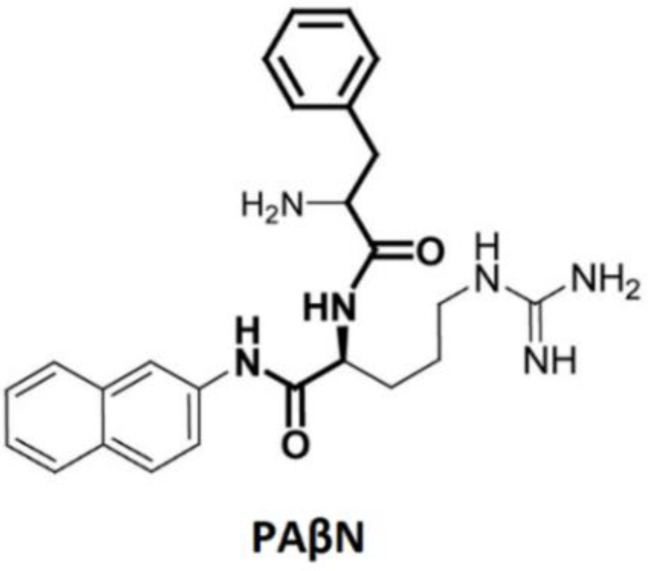
Chemical structure of PAβN.

**Figure 7 biology-11-01328-f007:**
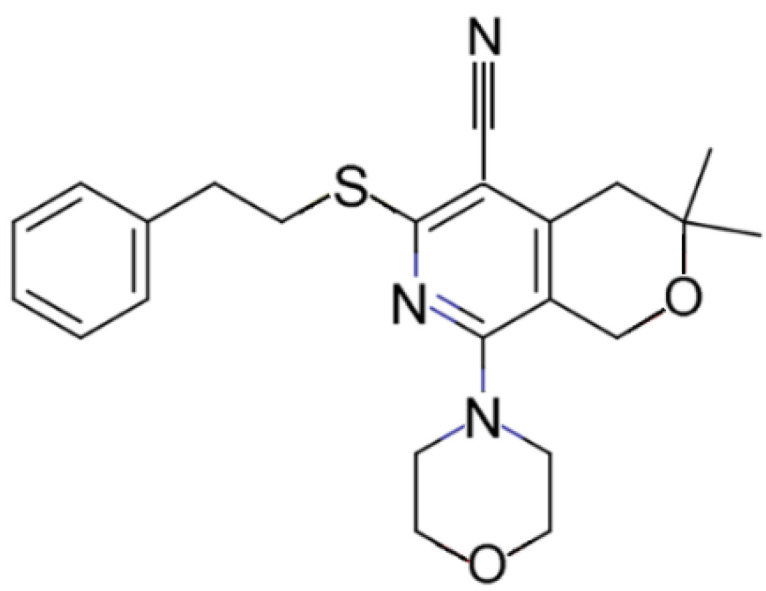
Chemical structure of MBX2319.

**Figure 8 biology-11-01328-f008:**
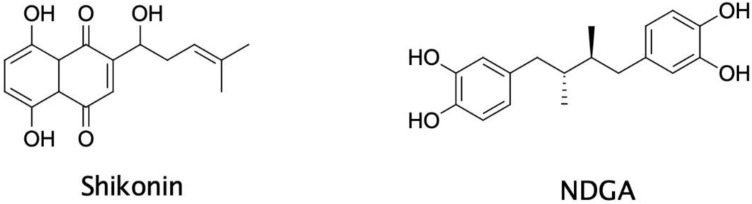
Chemical structures of the phytochemicals shikonin and NDGA.

**Figure 9 biology-11-01328-f009:**
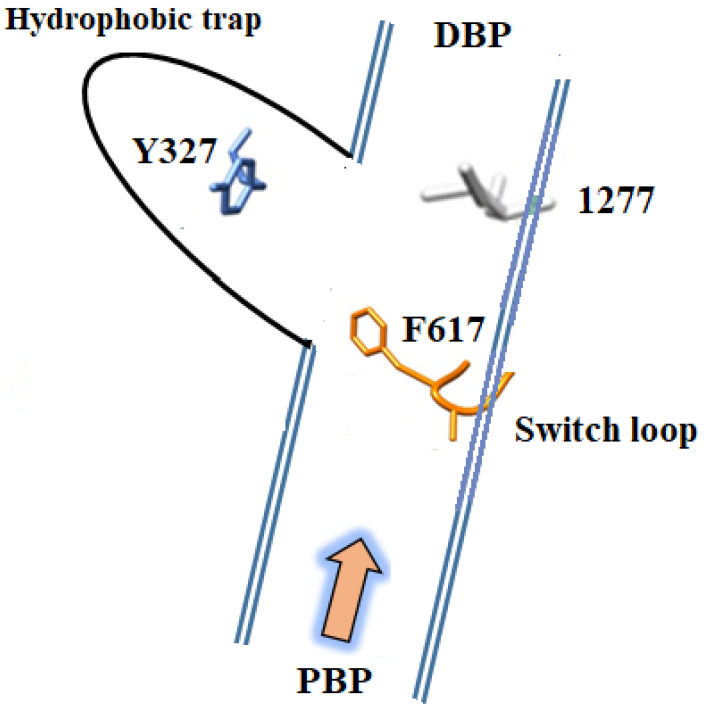
Schematic diagram showing the position of the hydrophobic trap relative to the proximal binding pocket (PBP) and the distal binding pocket (DBP). The switch loop that separates the PBP and DBP is shown in orange. Y327 which lies deep in the trap is highlighted, as is I277, which resides in the translocation tunnel. The yellow arrow depicts the direction substrates move through AcrB.

**Table 1 biology-11-01328-t001:** Examples of EPIs from non-plant sources against Gram-negative bacteria.

EPI Type	Bacteria	Efflux Substrate	References
Peptidomimetics	*E. coli, P. aeruginosa, K. pneumoniae, S. enterica, Campylobacter* spp., *E. aerogenes, A. baumanii*	CAM, FQs, macrolides, CAR, TETs	[66,73,74,75,76,77,78,79]
Arylpiperazines	*A. baumanii, P. aeruginosa, C. jejuni, Enterobacteriaceae (not Serratia)*	CAM, TETs, FQs, macrolides, LZD	[74,80,81,82,83,84]
Arylpiperidines	*E. coli*	LZD	[85]
Quinoline derivatives	*E. aerogenes, K. pneumoniae*	CAM, TET, NOR	[86,87,88,89,90]
Quinazoline derivatives	*E. aerogenes, P. aeruginosa,*	CAM, NAL, SPX	[91]
Phenothiazines	*E. coli, S. enterica, B. pseudomallei*	CAM, TETs, NAL, LVX, triclosan, ERY, aminoglycosides	[92,93,94]
Hydantoins	*E. coli, E. aerogenes*	CAM, NAL, SPX, doxycycline, ERY	[95,96]
Antibiotics globomycin	*E. aerogenes*	CAM, NOR	[76]
Trimethoprim	*Enterobacteriaceae, P. aeruginosa*	CAM, TET, CIP, ERY	[97]
Antibiotic analogs, Tetracycline analogs	*E. coli*	TETs	[98,99]
Fluoroquinolone analogs	*E. coli, P. aeruginosa*	FQs, macrolides	[100]
Aminoglycoside analogs	*H. influenza*	TET, GEN	[100]
Indole derivatives	*E. coli*	CAM, ERY, CIP, TET	[101]
Substituted polyamines	*H. influenza*	–	[102]
Pyranopyridines	*Enterobacteriaceae*	FQs, PIP	[9,44,70,103]
Pyridopyrimidines	*P. aeruginosa*	FQs, β-lactams	[104,105]
sRNA and antisense oligonucleotides	*E. coli, C. jejuni*	CIP, ERY	[100,106]
Microbial EPIs	*P. aeruginosa*	LVX	[107]
Serum compounds	*A. baumanii, P. aeruginosa*	MIN, CIP	[108]
Epinephrine	*Enterobacteriaceae, P. aeruginosa*	CAM, TET, CIP, ERY	[97]
Naphthamides	*E. coli*	CAM, TPP, ERY	[11,109]

CAM = chloramphenicol, CIP = ciprofloxacin, ERY = erythromycin, FQs = fluoroquinolones, GEN = gentamicin, LEV = levofloxacin, LZD = linezolid, NAL = nalidixic acid, NOR = norfloxacin, PIP = piperacillin, SPX = sparfloxacin, TET = tetracycline, TETs = tetracyclines, TPP = tetraphenylphosphonium.

**Table 2 biology-11-01328-t002:** Examples of EPIs from plants with inhibitory activity against Gram-negative bacteria efflux pumps.

Extract Compound	Plant	Bacteria Affected	References
Plumbagin	*Plumbago indica*	*E. coli*	[12]
Nordihydroguaretic acid	*Larrea tridentata*		
Shikonin	*Lithospermum erythrorhizon*		
Lysergol	*Ipomea muricata*		[118]
4-Hydroxy-α-tetralone + semisynthetic derivatives	*Ammannia* spp.		[119]
Ethanolic extract	*Baccharoides adoensis, Callistemon citrinus*	*Pseudomonas aeruginosa*	[120]
Lanatoside C	*Digitalis lanata*	*Pseudomonas aeruginosa, E. coli*	[121]
Ursolic acid	*Eucalyptus tereticornis*	*E. coli*	[122]
Daidzein	*Glycine max*		
Phenolic-rich maple syrup extracts (PRMSE)	*Acer saccharum*	*E. coli* *, Proteus mirabilis, Pseudomonas aeruginosa*	[123]
(−)-α-Pinene	*Alpinia katsumadai*	*Campylobacter jejuni*	[124]
Berberine, palmatine	*Berveris bulgaris*	*Pseudomonas aeruginosa*	[125]
Conessine	*Holarrhena antidysenterica*		[126]

## Data Availability

Not applicable.

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
