# Peer review of "Drug Efflux Pump Inhibitors: A Promising Approach to Counter Multidrug Resistance in Gram-Negative Pathogens by Targeting AcrB Protein from AcrAB-TolC Multidrug Efflux Pump from Escherichia coli"

_biology, 2022, doi:10.3390/biology11091328_

Round 1

Reviewer 1 Report

This manuscript is a review about RND multidrug efflux pumps, major contributors of intrinsic multidrug drug resistances in Gram-negative bacteria. The topic is very interesting but often reviewed in many journals for some decades. Unfortunately this manuscript features nothing new compared to already published reviews. In addition documentation itself is poor. 

Author Response

The topic was reviewed in many journals, but in this manuscript, the problem of multidrug-resistant was specifically discussed in Gram-negative bacteria, starting with the characteristics of Gram-negative bacteria, the difficulty of treating, the mechanisms of resistance, focusing on the most important mechanism, which is drug efflux pumps, especially the RND superfamily, and targeting the subunit protein AcrB pump because of its clinical importance also its analogues in many of Gram-negative bacteria.

The manuscript showed some graphics that explain and make it easier for novice readers to understand the topic better.  The chemical composition of AcrAB-TolC, especially AcrB, and the movement of compounds within it, were explained with the mention of some examples of inhibitors that are related to it, especially of plant origin, in order to avoid potential toxicity, while mentioning the importance of the hydrophobic trap and its possible role in finding safer clinical inhibitors with specific mechanisms of action. Moreover, the manuscript recommended the importance of designing inhibitors in the future so that they are suitable in terms of length and size to discourage the trap and prevent its action.

Based on the above, I think that the manuscript provided a full and distinct explanation from some previous research in determining the problem in general and then gradually determining how to treat it through an adequate explanation of the chemical composition of the efflux pumps and the identification of the target part of the inhibition, which is the hydrophobic trap with some suggestions for future work which will be contributed to the effective development of small molecule efflux pump inhibitors that target AcrB using a structure-guided design.

Reviewer 2 Report

The author sequentially well-described the MDR, efflux pumps, RND pumps, AcrAB-TolC pumps, structure and function of AcrB, and EPI targeting AcrB. This review encouraged the readers to understand basic knowledge and essentials of EPIs. Therefore, the manuscript can be published in the international journal. The introductory parts, subtitle 1 to 12, were well-organized and well-written to understand the basic and essential knowledge of efflux pumps and EPIs. However, I think that part 13 and 14 give very few information of EPIs. The author gave many examples of EPIs via Table 2 and 3, but the author exampled very few EPIs in the text. In Table 2 and 3, the authors should add the action mode or action site of EPIs. Finally, the author should add the possibility of clinical application of phytochemical EPIs.         

Specific comments

Line 132-152. The various types of efflux pumps should be described in this section.  

Line 70. Delete the term ‘microbiome’.

Line 99. Revise the subtitle: Gram-negative bacteria and difficulty in treatment

Line 150-151. ‘E. coli’ should be changed to Italic characters.

Line 157. Delete ‘many’.

Line 193-195. The abbreviations of IMP and PAP were previously used and, therefore, the full name of these two terms should be deleted.

Line 399. Please correct the typographical error of ‘deccreased’.  

Author Response

The author sequentially well-described the MDR, efflux pumps, RND pumps, AcrAB-TolC pumps, structure and function of AcrB, and EPI targeting AcrB. This review encouraged the readers to understand basic knowledge and essentials of EPIs. Therefore, the manuscript can be published in the international journal. The introductory parts, subtitle 1 to 12, were well-organized and well-written to understand the basic and essential knowledge of efflux pumps and EPIs.

However, I think that part 13 and 14 give very few information of EPIs.

"I will add more information about EPIs soon"

The author gave many examples of EPIs via Table 2 and 3, but the author exampled very few EPIs in the text.

"In tables, I gave diverse examples of EPIs of different types of bacteria, but in the text, I mentioned only new EPIs of botanical origin that specifically targeted AcrB which showed synergistic activity with a panel of antibiotics, not disrupted the bacterial inner or outer membranes, with similar potency with the peptidomimetic PAβN. Importantly, these EPIs did inhibit the AcrAB-TolC drug efflux pump in whole cell substrate transport assays."

In Table 2 and 3, the authors should add the action mode or action site of EPIs.

"I think that it is not necessary to mention the action mode or action site of EPIs, especially since the inhibitors mentioned in the tables were inhibitors of other types of bacteria and not specific to the bacteria targeted in the topic, but in the text, I mentioned the action mode or action site of EPIs that specifically targeted on AcrB."

Finally, the author should add the possibility of clinical application of phytochemical EPIs.        

"I will add more information about the possibility of using EPIs clinically soon".

Specific comments

Line 132-152. The various types of efflux pumps should be described in this section.

"I have done just like you asked."

Line 70. Delete the term ‘microbiome’.

"I have done just like you asked."

Line 99. Revise the subtitle: Gram-negative bacteria and difficulty in treatment

"I have done just like you asked."

Line 150-151. ‘E. coli’ should be changed to Italic characters.

"I have done just like you asked."

Line 157. Delete ‘many’.

"I have done just like you asked."

Line 193-195. The abbreviations of IMP and PAP were previously used and, therefore, the full name of these two terms should be deleted.

"I have done just like you asked."

Line 399. Please correct the typographical error of ‘deccreased’. 

"I have done just like you asked."

Reviewer 3 Report

In this work Rawaf Alenazy focus on the RND efflux pump AcrAB-TolC from E. coli. Topic is important and I understand why author decided to prepare that review, which might be very informative after gathering all of the information in this particular subject. However, the way of introduction of this topic require a lot of changes in case of this manuscript. Introduction is too long, information are repeated through the manuscript a lot, a lot of times. It should be definitely condensed, better organized, and should build story from the general to the specific parts of topic. Some specific comments below:

In title use Escherichia coli, as well as in a manuscript when it is appearing for the first time.

Line 4: two times: AcrAB-TolC AcrAB-TolC

Line 14: E. coli in italics: check in the rest of manuscript

Line 31:  multidrug resistance (MDR) abbreviation already appeared in a line 27

Lines 68-70: manuscript require language revision, as it is sometimes difficult to understand

Line 75: “MDR infections, in particular by Gram-negative pathogens, are becoming a worldwide health problem that needs new and effective solutions (Venter, 2019).” It is almost repeating of sentence from lines 70-75, lines 91-93, 27.   I have impression I am reading the same things all around. There is no purpose of Paragraph 3, short note about structure differences can be implemented here in one sentence. Eg. Due to multi-layered envelope, that is giving them ability to control the permeability of the harmful substances in comparison to Gram-positive bacteria, which possess only single cell membrane.

From Paragraph 2 to line 119 there are a lot of information that are repeated,. Eg. “ These multiple membranes confer the Gram-negative bacteria the ability to control the permeability of the harmful substances entering insidethe bacteria such as toxic substances and antibiotics (Du et al., 2018; Blair et al., 2014; 106 Venter et al., 2015; Nikaido, 2011)” and next sentence: ‘Moreover, these membranes confer the Gram-negative bacteria a fundamental resistance against many antibiotics of different chemical compositions (Exner et al., 2017).” It can be one sentence : “ These multiple membranes confer the Gram-negative  bacteria the ability to control the permeability of the harmful substances entering inside  the bacteria such as toxic substances and antibiotics with different chemical composition”.

Paragraph 4: Line 114: “In pathogenic bacteria, there are many mechanisms can confer MDR to bacteria against several different class of antibiotics”, line 118: ‘which in turn developed their mechanisms in order to eliminate the harmful effect of antibiotics”, line 119: “Pathogenic bacteria generally have mechanisms of antibiotic resistance which include”. Instead of repeating the same statements Author should show exact examples, or focus more on a subject that is in title: “A promising approach to Counter Multidrug Resistance in Gram-Negative Pathogens by target-3 ing AcrB protein from AcrAB-TolC AcrAB-TolC multidrug efflux pump from E. coli”. This all until line 153 should be a short introduction for 1-1.5 sites maximum. Not 5 paragraphs of introduction as paragraph 5 ends with: Hence, this review will specifically focus on the RND efflux pump AcrAB-TolC from E. coli…

Line 142: Moreovere, efflux pumps have essential role in causing virulence, pathogenesis and biofilm formation. Next sentence: The role of efflux pumps is not limited to the above, they have importnt role as well in bacterial pathogenicity in various ways. It makes no sense.

Lines 170-172: The  RND superfamily drug transporters consist of three proteins; an outer-membrane protein (OMP), a periplasmic adapter protein (PAP), which connects IMP with OMP and an inner-membrane protein (IMP) (Figure 1). Please make some proper order, it looks like: connects IMP with OMP and IMP ??

Line 164: RND transporters capture this wide variety of substrates from the cytoplasm, the outer leaflet of the inner membrane or the periplasm. Line 173: These proteins work as tripartite pumps and their function is to recognize and bind substrates from the cytoplasm, the outer leaflet of the inner membrane or the periplasm.

Quality of Figure 1 should be improved.

Line 154: The RND efflux pump family, to which AcrAB-TolC belongs, line 189: The E. coli AcrAB-TolC drug efflux complex, a member of the RND superfamily. The same information is repeated again through manuscript.

Line 193: Abbreviations of IMP, PAP were already introduced in manuscript.

Line 353: it is Figure 3 not 7.

References: please organize references, once doi is included, once not.

Author Response

In this work Rawaf Alenazy focus on the RND efflux pump AcrAB-TolC from E. coli. Topic is important and I understand why author decided to prepare that review, which might be very informative after gathering all of the information in this particular subject. However, the way of introduction of this topic require a lot of changes in case of this manuscript. Introduction is too long, information are repeated through the manuscript a lot, a lot of times. It should be definitely condensed, better organized, and should build story from the general to the specific parts of topic.

I have changed all the comments mentioned above such as minimising the introduction, and deleting the repeated sentences.

Some specific comments below:

In title use Escherichia coli, as well as in a manuscript when it is appearing for the first time.

"I have done the change just like you asked."

Line 4: two times: AcrAB-TolC AcrAB-TolC

"I have deleted the repeating word."

Line 14: E. coli in italics: check in the rest of manuscript

"I have done the change just like you asked."

Line 31:  multidrug resistance (MDR) abbreviation already appeared in a line 27

"I have done the change just like you asked."

Lines 68-70: manuscript require language revision, as it is sometimes difficult to understand

"I have improved the sentences to be more clear just like you asked."

Line 75: “MDR infections, in particular by Gram-negative pathogens, are becoming a worldwide health problem that needs new and effective solutions (Venter, 2019).” It is almost repeating of sentence from lines 70-75, lines 91-93, 27.   I have impression I am reading the same things all around.

"I have done the change which is deleting the repeating information."

There is no purpose of Paragraph 3, short note about structure differences can be implemented here in one sentence. Eg. Due to multi-layered envelope, that is giving them ability to control the permeability of the harmful substances in comparison to Gram-positive bacteria, which possess only single cell membrane.

"I have done the changes to make the sentences more consistent with the idea of the topic."

From Paragraph 2 to line 119 there are a lot of information that are repeated,. Eg. “ These multiple membranes confer the Gram-negative bacteria the ability to control the permeability of the harmful substances entering insidethe bacteria such as toxic substances and antibiotics (Du et al., 2018; Blair et al., 2014; 106 Venter et al., 2015; Nikaido, 2011)” and next sentence: ‘Moreover, these membranes confer the Gram-negative bacteria a fundamental resistance against many antibiotics of different chemical compositions (Exner et al., 2017).” It can be one sentence : “ These multiple membranes confer the Gram-negative  bacteria the ability to control the permeability of the harmful substances entering inside  the bacteria such as toxic substances and antibiotics with different chemical composition”.

"I have done the changes which is deleting the repeating information and making the sentences more consistent with the idea of the topic."

Paragraph 4: Line 114: “In pathogenic bacteria, there are many mechanisms can confer MDR to bacteria against several different class of antibiotics”, line 118: ‘which in turn developed their mechanisms in order to eliminate the harmful effect of antibiotics”, line 119: “Pathogenic bacteria generally have mechanisms of antibiotic resistance which include”. Instead of repeating the same statements Author should show exact examples, or focus more on a subject that is in title: “A promising approach to Counter Multidrug Resistance in Gram-Negative Pathogens by target-3 ing AcrB protein from AcrAB-TolC AcrAB-TolC multidrug efflux pump from E. coli”. This all until line 153 should be a short introduction for 1-1.5 sites maximum. Not 5 paragraphs of introduction as paragraph 5 ends with: Hence, this review will specifically focus on the RND efflux pump AcrAB-TolC from E. coli…

"I have done the change which is to minimize the introduction information."

Line 142: Moreovere, efflux pumps have essential role in causing virulence, pathogenesis and biofilm formation. Next sentence: The role of efflux pumps is not limited to the above, they have importnt role as well in bacterial pathogenicity in various ways. It makes no sense.

"I have changed and improved the sentences to be more consistent"

Lines 170-172: The RND superfamily drug transporters consist of three proteins; an outer-membrane protein (OMP), a periplasmic adapter protein (PAP), which connects IMP with OMP and an inner-membrane protein (IMP) (Figure 1). Please make some proper order, it looks like: connects IMP with OMP and IMP ??

"I have done the change which is making proper order."

Line 164: RND transporters capture this wide variety of substrates from the cytoplasm, the outer leaflet of the inner membrane or the periplasm.

"I have done the change which is deleting the repeating information just like you asked."

Line 173: These proteins work as tripartite pumps and their function is to recognize and bind substrates from the cytoplasm, the outer leaflet of the inner membrane or the periplasm.

"I have done the change which is deleting the repeating information just like you asked."

Quality of Figure 1 should be improved.

I have modified the Figure to be more consistent

Line 154: The RND efflux pump family, to which AcrAB-TolC belongs,

"I have done the change just like you asked."

line 189: The E. coli AcrAB-TolC drug efflux complex, a member of the RND superfamily. The same information is repeated again through manuscript.

"I have done the change which is deleting the repeating information just like you asked."

Line 193: Abbreviations of IMP, PAP were already introduced in manuscript.

"I have done the change just like you asked."

Line 353: it is Figure 3 not 7.

"I have done the change just like you asked."

References: please organize references, once doi is included, once not.

"I have done the change just like you asked."

Reviewer 4 Report

Analysis by working partitions:

1 - Introduction: must be reformed in the content and in the writing of the general part review the syntax of the topic

2- Discussion: deepen the discussion and given the shortage of antibiotic drugs against multidrug-resistant strains and in particular against Gram-negative, the use of pump inhibitors with synergistic activity with antibiotics in therapy. Learn more about this by using and citing the following references:

PMID: 32036352 ; PMID: 29412107  ; PMID: 29424299

3 - Check the bibliographic entries in the text, some of which are non-compliant, review some entries in the bibliographic references and necessarily insert those referred to in point 2 for the purpose of my acceptance.

4 - Review the English grammar and in particular the applied scientific English: in particular, verbal tenses and syntax in the discussion.

Author Response

Analysis by working partitions:

1 - Introduction: must be reformed in the content and in the writing of the general part review the syntax of the topic

"I have done just like you asked."

2- Discussion: deepen the discussion and given the shortage of antibiotic drugs against multidrug-resistant strains and in particular against Gram-negative, the use of pump inhibitors with synergistic activity with antibiotics in therapy. Learn more about this by using and citing the following references:

PMID: 32036352 ; PMID: 29412107  ; PMID: 29424299

"I will add more information that deeply discusses the importance of the inhibitors of drug efflux pumps in the presence of the high shortage in the drugs of multidrug-resistant strains and in particular against Gram-negative"

3 - Check the bibliographic entries in the text, some of which are non-compliant, review some entries in the bibliographic references and necessarily insert those referred to in point 2 for the purpose of my acceptance.

"I have done just like you asked."

4 - Review the English grammar and in particular the applied scientific English: in particular, verbal tenses and syntax in the discussion.

"I have done just like you asked."